# VHEGAN: Variational Hetero-Encoder Randomized GAN for Zero-Shot Learning

## Abstract

To extract and relate visual and linguistic concepts from images and textual descriptions for text-based zero-shot learning (ZSL), we develop variational hetero-encoder (VHE) that decodes text via a deep probabilisitic topic model, the variational posterior of whose local latent variables is encoded from an image via a Weibull distribution based inference network. To further improve VHE and add an image generator, we propose VHE randomized generative adversarial net (VHE-GAN) that exploits the synergy between VHE and GAN through their shared latent space. After training with a hybrid stochastic-gradient MCMC/variational inference/stochastic gradient descent inference algorithm, VHEGAN can be used in a variety of settings, such as text generation/retrieval conditioning on an image, image generation/retrieval conditioning on a document/image, and generation of text-image pairs. The efficacy of VHEGAN is demonstrated quantitatively with experiments on both conventional and generalized ZSL tasks, and qualitatively on (conditional) image and/or text generation/retrieval.

## 1 Introduction

There has been significant recent interest in zero-shot learning (ZSL) that leverages auxiliary semantic information to transfer the knowledge learned from the training categories (seen classes) to testing ones (unseen classes) (Fu et al., 2018; Elhoseiny et al., 2017a). A particularly challenging task is text-based ZSL (Elhoseiny et al., 2013; 2017a; Ba et al., 2015; Elhoseiny et al., 2017b; Qiao et al., 2016; Zhu et al., 2018), which assumes that there are $S$ seen image classes and $U$ unseen ones, and image class $c \in \{1, \ldots, S + U\}$ is associated with both a set of $N_c$ images $\{x_{cn}\}_{n=1,N_c}$ and a textual description $t_c$. In the training stage, all or a subset of the images and textual descriptions of the $S$ seen classes, denoted as $\{\{(x_{cn}, t_c)\}_{n=1,N_c}\}_{c=1,S}$, are used to learn the model. While in the testing stage, one removes the class label of an image from the unseen classes, and then maps that image to the text description of one of the $U$ unseen classes; if the true mapping is the same as the most likely mapping (or among the top-five mappings) ranked by a ZSL algorithm, it is considered as a correct classification in terms of Top-1 accuracy (or Top-5 accuracy). The average Top-1 (or Top-5) accuracy of these mappings is used to evaluate the performance of a ZSL algorithm. One may also consider a generalized ZSL (GZSL) setting, at the testing stage of which each held-out image, from a seen/unseen class that is not used for training, needs to be mapped to one of the $S + U$ classes (Socher et al., 2013; Chao et al., 2016; Verma et al., 2018).

Given the success of deep generative models in unraveling complex data structure (Kingma & Welling, 2014; Kingma et al., 2014; Zhou et al., 2016), we are motivated to apply them to ZSL, especially when there are not many labeled data for the seen classes (Wang et al., 2018b; Verma et al., 2018). *Although some image caption models (Wang et al., 2017; Xu et al., 2015) are able to exploit sequential information to related an image to a simple sentence, and a sequential description could be excellent at defining a specific individual image, the key words could be more effective to define a class of images (i.e., not a single image). In addition, the text description for a class of images could vary from just a few words (e.g., a sentence or several tags) to hundreds of sentences (e.g., a document), whose key words relevant for ZSL could be more robustly identified under the bag-of-words (BOW) text representation. For these reasons, to address the challenges of text-based ZSL*, we first introduce a variational hetero-encoder (VHE) that encodes an image to decode its *BOW* textual description. The proposed VHE is related to a usual variational auto-encoder (VAE) (Kingma & Welling, 2014; Rezende et al., 2014) in using variational inference (Jordan et al., 1999;

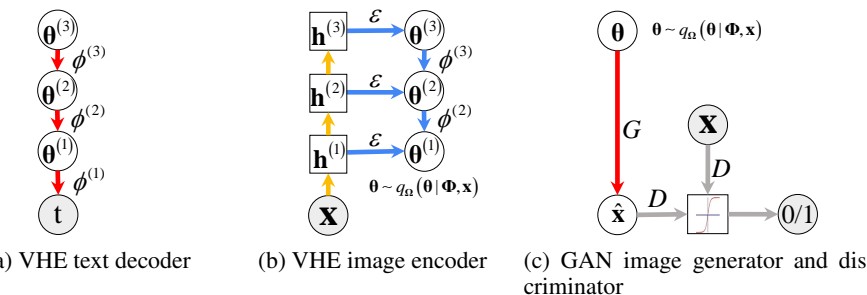

(a) VHE text decoder     (b) VHE image encoder     (c) GAN image generator and discriminator

Figure 1: The overall model architecture of variational hetero-encoder (VHE), consisting of (a) and (b), and VHE randomized GAN (VHEGAN), consisting of (a), (b), and (c), where (a) is the VHE text decoder, a three-hidden-layer Poisson gamma belief network, (b) is the VHE image encoder, a Weibull distribution based deterministic-upward (denoted with orange arrows) and stochastic-downward (denoted with blue arrows) variational encoder, and (c) is the GAN image generator and discriminator of VHEGAN.

Blei et al., 2017; Hoffman et al., 2013) to jointly optimize its encoder and decoder, but differs from a VAE in making the input of the encoder, which is an image, different from the output of the decoder, which is a textual description associated with that image. *Generative adversarial nets (GANs) (Goodfellow et al., 2014) and their various extensions, such as deep convolutional GANs Radford et al. (2016), have made significant progresses in generating high-quality synthetic images. To further explore tighter relationships between images and texts to improve the performance of VHE for ZSL and add the capability to generate random images conditioning on an image, a document, or random noise,* we further propose **V**ariational **H**etero-**E**ncoder randomized **G**enerative **A**dversarial **N**et (VHEGAN) that exploits the synergy between VHE and GAN.

VHEGAN is distinct in several ways from the original GAN (Goodfellow et al., 2014), which does not have an encoder, has little control over its latent space and hence the characteristics of generated images, and imposes no dependency relationship between the real images fed to its discriminator and the random noise fed to its generator. By contrast, VHEGAN inherits both the VHE image encoder and text decoder, makes the random noise fed into the GAN image generator be dependent on the image fed into the GAN discriminator during training, and can condition its latent space on either a document or an image during testing. Consequently, VHEGAN is capable of performing four different tasks: 1) text to image: generating images with the GAN image generator, or retrieving images, given the conditional posterior of a document under the VHE text decoder; 2) image to text: generating documents with the VHE text decoder, or retrieving documents, given the variational posterior produced by the VHE image encoder fed with a natural image (text-ZSL is such an task); 3) image to image: generating images with the GAN image generator, or retrieving images, given the variational posterior produced by the VHE image encoder fed with a natural image, with the similarity between a generated image and the input one measurable by the GAN discriminator; and 4) noise to text-image pair: generating a document using the VHE text decoder and a corresponding image by feeding the latent variable of the VHE text decoder into the GAN image generator.

## 2 JOINT TEXT AND IMAGE GENERATION FOR ZERO-SHOT LEARNING

The proposed VHE uses the Poisson gamma belief network (GBN) of Zhou et al. (2015) as its text decoder, whose output is a bag-of-words count vector, as shown in Fig. 1 (a), and a Weibull distribution based deterministic-upward–stochastic-downward inference network (Zhang et al., 2018) as its image encoder, whose input is a natural image, as shown in Fig. 1 (b). Integrating VHE and GAN, the proposed VHEGAN, as shown in Fig. 1 (c), tightly couples the VHE text generator, VHE image encoder, and GAN image generator and discriminator by imposing the variational posterior of the bag-of-words vector of a document, under the Poisson GBN text decoder, be the distribution of the random noise fed into the GAN image generator; the variational posterior is encoded from an image associated with that document via the Weibull distribution based image encoder; and the GAN discriminator tries to discriminate between that selected image and the generated one. All the three components that are structurally coupled with each other will be jointly optimized.

## 2.1 VHE Text decoder (generator): Poisson gamma belief network

*Two methods are popular in representing texts: word embedding (Bengio et al., 2003; Mikolov et al., 2013) and BOW count vectors. Considering the need of semantically relate a class of images to the key words in the corresponding class-specific textual description,* we represent the text descriptions as BOW count vectors as $\left\{ \boldsymbol{t}_c \in \mathbb{Z}^{K_0} \right\}_{c=1}^{S+U}$, where $\mathbb{Z} = \{0, 1, \cdots\}$ and $K_0$ is the vocabulary size. To extract hierarchical latent representation from these high-dimensional sparse count vectors $\boldsymbol{t}_c$, we choose the Poisson GBN (Zhou et al., 2015), which can be considered as a member of deep exponential families (Ranganath et al., 2015). It consists of multiple gamma distributed stochastic hidden layers, generalizing the "shallow" Poisson factor analysis (Zhou et al., 2012; Zhou & Carin, 2015) that is closely related to latent Dirichlet allocation (LDA) (Blei et al., 2003) into a deep setting. More specifically, modeling $\boldsymbol{t}_c$ under the Poisson likelihood, its generative model with $L$ hidden layers, from top to bottom, is expressed as

$$\boldsymbol{\theta}_c^{(L)} \sim \text{Gam}\left(\boldsymbol{r}, 1/s_c^{(L+1)}\right), \ldots, \boldsymbol{\theta}_c^{(l)} \sim \text{Gam}\left(\boldsymbol{\Phi}^{(l+1)}\boldsymbol{\theta}_c^{(l+1)}, 1/s_c^{(l+1)}\right), \ldots,$$

$$\boldsymbol{\theta}_c^{(1)} \sim \text{Gam}\left(\boldsymbol{\Phi}^{(2)}\boldsymbol{\theta}_c^{(2)}, 1/s_c^{(2)}\right), \ \boldsymbol{t}_c \sim \text{Pois}\left(\boldsymbol{\Phi}^{(1)}\boldsymbol{\theta}_c^{(1)}\right), \tag{1}$$

where the hidden units $\boldsymbol{\theta}_c^{(l)} \in \mathbb{R}_+^{K_l}$ of layer $l$ are factorized under the gamma likelihood into the product of the factor loading $\boldsymbol{\Phi}^{(l)} \in \mathbb{R}_+^{K_{l-1} \times K_l}$ and hidden units of the next layer, $\mathbb{R}_+ = \{x, x \geq 0\}$, $s_c^{(l)} > 0$, and $K_l$ is the number of topics (factors) of layer $l$. We further restrict that the sum of each column of $\boldsymbol{\Phi}^{(l)}$ is equal to one via a Dirichlet prior. The topics $\boldsymbol{\Phi}^{(l)}$ at hidden layer $l$ can be visualized as $\left[\prod_{t=1}^{l-1}\boldsymbol{\Phi}^{(t)}\right]\boldsymbol{\phi}_k^{(l)}$, which are found to be very specific in the bottom layer and becoming increasingly more general when moving upwards. More details about the Poisson GBN can be found in Zhou et al. (2016). Note as shown in Cong et al. (2017), the Poisson GBN can also be represented as deep LDA, an alternative representation that facilitates the derivation of a scalable stochastic-gradient MCMC algorithm that is well suited for deep topic modeling. For simplicity, below we use the Poisson GBN to refer to both the Poisson GBN and deep LDA representations of the same underlying deep generative model.

Choosing a deep generative model as the text decoder distinguishes VHE from Gomez et al. (2017) that uses LDA (Blei et al., 2003), a "shallow" probabilistic model, and Ba et al. (2015) that uses non-probabilistic black-box DNNs to learn class-specific document features.

## 2.2 VHE image encoder: Weibull upward-downward variational encoder

The Poisson GBN is equipped with both a closed-form upward-downward Gibbs sampler that is suitable for a moderate-sized corpus (Zhou et al., 2016), and a topic-layer-adaptive stochastic gradient Riemannian (TLASGR) MCMC that is scalable to a big corpus (Cong et al., 2017). To avoid using iterations to sample the latent representation of a testing document from its posterior, Zhang et al. (2018) develop Weibull upward-downward variational encoder (WUDVE) that directly projects an observed bag-of-words vector into its posterior distribution given a random sample of the global parameters, leading to extremely fast inference for out-of-sample prediction.

Let us denote $\boldsymbol{\Phi} = \{\boldsymbol{\Phi}^{(1)}, \ldots, \boldsymbol{\Phi}^{(L)}, \boldsymbol{r}\}$, where $\boldsymbol{\Phi}^{(l)} = \{\boldsymbol{\phi}_k^{(l)}\}_{k,l}$, as the set of global parameters of the Poisson GBN shown in (1). In this paper, we leverage WUDVE as the variational inference network to approximate the true posterior distribution $p(\boldsymbol{\theta} \mid \boldsymbol{t}, \boldsymbol{\Phi})$, where $\boldsymbol{\theta} = \{\boldsymbol{\theta}^{(l)}\}_{1,L}$ denotes the set of local parameters, but change the input of WUDVE from the bag-of-words vector $\boldsymbol{t}$ into an image $\boldsymbol{x}$ associated with $\boldsymbol{t}$. More specifically, as illustrated in Fig. 1b, we construct it as

$$q_{\boldsymbol{\Omega}}(\boldsymbol{\theta} \mid \boldsymbol{\Phi}, \boldsymbol{x}) := \left[\prod_{l=1}^{L-1} q_{\boldsymbol{\Omega}}(\boldsymbol{\theta}^{(l)} \mid \boldsymbol{\Phi}^{(l+1)}, \boldsymbol{\theta}^{(l+1)}, \boldsymbol{x})\right] q_{\boldsymbol{\Omega}}(\boldsymbol{\theta}^{(L)} \mid \boldsymbol{r}, \boldsymbol{x}), \tag{2}$$

where, with $f(\boldsymbol{x})$ representing a feature vector deterministically projected from $\boldsymbol{x}$ via an off-the-shelf pre-trained deep neural network $f(\cdot)$, we define

$$q_{\boldsymbol{\Omega}}(\boldsymbol{\theta}^{(l)} \mid \boldsymbol{\Phi}^{(l+1)}, \boldsymbol{\theta}^{(l+1)}, \boldsymbol{x}) = \text{Weibull}(\boldsymbol{k}_{\boldsymbol{\Omega}}^{(l)}(f(\boldsymbol{x})) + \boldsymbol{\Phi}^{(l+1)}\boldsymbol{\theta}^{(l+1)}, \boldsymbol{\lambda}_{\boldsymbol{\Omega}}^{(l)}(f(\boldsymbol{x}))),$$

$$q_{\boldsymbol{\Omega}}(\boldsymbol{\theta}^{(L)} \mid \boldsymbol{r}, \boldsymbol{x}) = \text{Weibull}(\boldsymbol{k}_{\boldsymbol{\Omega}}^{(L)}(f(\boldsymbol{x})) + \boldsymbol{r}, \boldsymbol{\lambda}_{\boldsymbol{\Omega}}^{(L)}(f(\boldsymbol{x}))), \tag{3}$$

where $k_{\boldsymbol{\Omega}}^{(l)}(f(\boldsymbol{x})) \in \mathbb{R}^{K_l}$ and $\boldsymbol{\lambda}_{\boldsymbol{\Omega}}^{(l)}(f(\boldsymbol{x})) \in \mathbb{R}^{K_l}$ are deterministically transformed from $f(\boldsymbol{x})$. We use $\boldsymbol{\Omega}$ to denote all the parameters of the inference network.

Let us denote $p_{data}(\boldsymbol{x}, \boldsymbol{t}) = \frac{1}{\sum_{c=1}^{C}\sum_{n=1}^{N_c}}\sum_{c=1}^{C}\sum_{n=1}^{N_c}\delta_{(\boldsymbol{x}_{cn}, \boldsymbol{t}_c)}$ as the empirical data distribution for the image-text pairs. Given $\boldsymbol{\Phi}$ of the Poisson GBN, to approximate the posterior distribution of the local parameters $\boldsymbol{\theta}$ given text description $\boldsymbol{t}$, we use WUVDE $q_{\boldsymbol{\Omega}}(\boldsymbol{\theta} \,|\, \boldsymbol{\Phi}, \boldsymbol{x})$, whose input is an image $\boldsymbol{x}$, and learn its parameters $\boldsymbol{\Omega}$ by maximizing the evidence lower bound (ELBO) as

$$\max_{\boldsymbol{\Omega}} L_{\text{VHE}}(\boldsymbol{\Omega}) = \mathbb{E}_{(\boldsymbol{x},\boldsymbol{t})\sim p_{data}(\boldsymbol{x},\boldsymbol{t})} \left\{ \mathbb{E}_{q_{\boldsymbol{\Omega}}(\boldsymbol{\theta}\,|\,\boldsymbol{\Phi},\boldsymbol{x})} \left[ \ln p\left(\boldsymbol{t}\,|\,\boldsymbol{\Phi}^{(1)},\boldsymbol{\theta}^{(1)}\right) - \ln \frac{q_{\boldsymbol{\Omega}}(\boldsymbol{\theta}\,|\,\boldsymbol{\Phi},\boldsymbol{x})}{p(\boldsymbol{\theta}\,|\,\boldsymbol{\Phi})} \right] \right\}, \quad (4)$$

where $p(\boldsymbol{\theta}\,|\,\boldsymbol{\Phi}) = \left[\prod_{l=1}^{L-1} p(\boldsymbol{\theta}^{(l)}\,|\,\boldsymbol{\Phi}^{(l+1)}, \boldsymbol{\theta}^{(l+1)}, s^{(l+1)})\right] p(\boldsymbol{\theta}^{(L)}\,|\,\boldsymbol{r}, s^{L+1})$ is the prior distribution.

We refer to Zhang et al. (2018) for more details on how WUDVE can be combined with TLASGR-MCMC (Cong et al., 2017) to develop Weibull hybrid autoencoding inference (WHAI), a hybrid SG-MCMC/variational inference that represents the posterior of $\boldsymbol{\Phi}$ using MCMC samples, and the posterior of $\boldsymbol{\theta}$ given a sample of $\boldsymbol{\Phi}$ using WUDVE. To further improve the proposed VHE, below we show how to integrate it with a modified GAN to construct VHEGAN.

## 2.3 Image generator: VHE randomized generative adversarial net

*Both VAE and GAN are widely used for image generation, but we find in experiments that VAE is not expressive enough for mid/high-resolution RGB images ($64 \times 64$ resolution in our task) to clearly help the learning of VHE, whereas GAN does better for this purpose. Thus, we combine GAN and VHE in a novel way via a shared latent space, to further explore the relationships between two different modes for ZSL.* The original GAN consists of both a generator $G$ and a discriminator $D$, whose parameters are learned by optimizing a mini-max objective function as

$$\min_{G} \max_{D} L_{\text{GAN}}(D, G) = \mathbb{E}_{\boldsymbol{x}\sim p_{data}(\boldsymbol{x})}\left[\ln D(\boldsymbol{x})\right] - \mathbb{E}_{\boldsymbol{z}\sim p(\boldsymbol{z})}\left[\ln D(G(\boldsymbol{z}))\right], \quad (5)$$

where $p(\boldsymbol{z})$ is a random noise distribution that acts as the source of randomness to generate images (Goodfellow et al., 2014). In this paper, to exploit the synergy between variational inference and generative adversarial learning to achieve improved performance for both VHE and GAN, we use the variational posterior of VHE given an image, which is the one fed into the GAN image discriminator, as the random noise distribution of the GAN image generator.

More specifically, we modify the GAN objective function as

$$\min_{G} \max_{D} L_{\text{GAN}}(D, G) = \min_{G} \max_{D} \mathbb{E}_{\boldsymbol{x}\sim p_{data}(\boldsymbol{x})}\left\{\ln D(\boldsymbol{x}) - \mathbb{E}_{\boldsymbol{\theta}\sim q_{\boldsymbol{\Omega}}(\boldsymbol{\theta}\,|\,\boldsymbol{\Phi},\boldsymbol{x})}\left[\ln D(G(\boldsymbol{\theta}))\right]\right\} \quad (6)$$

where $q_{\boldsymbol{\Omega}}(\boldsymbol{\theta}\,|\,\boldsymbol{\Phi}, \boldsymbol{x})$ is defined as in (2). To jointly optimize the VHE and GAN components of VHEGAN, we merge the expectations in (4) and (6) to define its loss function as

$$\min_{G,\boldsymbol{\Omega}} \max_{D} \mathbb{E}_{(\boldsymbol{x},\boldsymbol{t})\sim p_{data}(\boldsymbol{x},\boldsymbol{t})} \left\{ \mathbb{E}_{q_{\boldsymbol{\Omega}}(\boldsymbol{\theta}\,|\,\boldsymbol{\Phi},\boldsymbol{x})} \left[ -\ln p\left(\boldsymbol{t}\,|\,\boldsymbol{\Phi}^{(1)},\boldsymbol{\theta}^{(1)}\right) + \ln \frac{q_{\boldsymbol{\Omega}}(\boldsymbol{\theta}\,|\,\boldsymbol{\Phi},\boldsymbol{x})}{p(\boldsymbol{\theta}\,|\,\boldsymbol{\Phi})} - \ln D(G(\boldsymbol{\theta})) \right] + \ln D(\boldsymbol{x}) \right\}. \quad (7)$$

It is important to note that the update of the VHE image encoder parameters $\boldsymbol{\Omega}$ is related to not only the ELBO of VHE, but also a modified GAN mini-max objective function, forcing the variational posterior $q_{\boldsymbol{\Omega}}(\boldsymbol{\theta}\,|\,\boldsymbol{\Phi}, \boldsymbol{x})$ to serve as a bridge between the image and text modalities.

For inference, since the Weibull distribution $\theta \sim \text{Weibull}(k, \lambda)$ can be reparameterized as $\theta = \lambda(-\ln(1-\epsilon))^{1/k}$, where $\epsilon \sim \text{Uniform}(0, 1)$, it is convenient to calculate the gradients of (7) with respect to $\{\boldsymbol{\Omega}, G, D\}$ by back-propagation (BP) using reparameterized random samples from $q_{\boldsymbol{\Omega}}(\boldsymbol{\theta}\,|\,\boldsymbol{\Phi}, \boldsymbol{x})$. The network architecture for the overall model shown in Fig. 1 is described in detail in the Appendix. What's more, by the aid of TLASGR-MCMC of Cong et al. (2017), we can collect posterior MCMC samples for the global parameters $\boldsymbol{\Phi}$ of the Poisson GBN, and develop a hybrid SG-MCMC/VHE/SGD end-to-end learning algorithm (Zhang et al., 2018) for the parameters of VHEGAN, as detailedly described in Algorithm 1 in the Appendix.

Note we choose the GAN generator and discriminator to be the same as those of deep convolutional GAN of Radford et al. (2016). Other more recent GANs, such as Wasserstein GAN (Arjovsky et al., 2017), Wasserstein GAN with gradient penalty (Gulrajani et al., 2017), spectral norm GAN (Miyato et al., 2018), and progressive growing GAN (Karras et al., 2018), may be combined with the VHE to construct even more expressive VHEGANs. We leave these extensions for future study.

## 2.4 ZERO-SHOT LEARNING AND GENERALIZED ZERO-SHOT LEARNING

The proposed VHE by itself and VHEGAN can both be used for ZSL in the same way: with the Poisson GBN based VHE text decoder and the WUDVE based VHE image encoder, to perform zero-shot classification for a test image $x$ from an unseen class, we predict its class label as

$$\underset{c\in\{S+1,\cdots,S+U\}}{\arg\max} \mathbb{E}_{\mathbf{\Phi}}\mathbb{E}_{\boldsymbol{\theta}\sim q_{\mathbf{\Omega}}(\boldsymbol{\theta}\,|\,\mathbf{\Phi},\boldsymbol{x})}[p(\boldsymbol{t}_c\,|\,\mathbf{\Phi},\boldsymbol{\theta})] \approx \underset{c\in\{S+1,\cdots,S+U\}}{\arg\max} \frac{1}{JK}\sum_{j=1}^{J}\sum_{k=1}^{K}p(\boldsymbol{t}_c\,|\,\mathbf{\Phi}_{(j)},\boldsymbol{\theta}_{(jk)}), \quad (8)$$

where $\{\mathbf{\Phi}_{(j)}\}_{j=1}^{J}$ is a set of $J$ collected posterior MCMC samples and $\boldsymbol{\theta}_{(j1)},\ldots,\boldsymbol{\theta}_{(jK)} \overset{iid}{\sim} q_{\mathbf{\Omega}}(\boldsymbol{\theta}\,|\,\mathbf{\Phi}_{(j)},\boldsymbol{x})$. We set $J$ and $K$ as 50 and 10, respectively, in our experiements. Note that VHE-GAN differs from VHE in that the optimization for the image encoder parameters $\mathbf{\Omega}$ is also influenced by the modified GAN min-max objective function, as shown in (7).

Conventional ZSL methods often make a restrictive assumption that a test example only comes from one of the unseen classes. A more challenging setting where a test example can come from either a seen class or an unseen one is known as generalized ZSL (GZSL) (Socher et al., 2013; Verma et al., 2018; Chao et al., 2016). To perform GZSL, one may introduce class-specific classifiers (Verma et al., 2018; Elhoseiny et al., 2017a;b). While neither the VHE or VHEGAN are equipped with class-specific classifiers, they can be directly applied for GSZL by simply changing $\arg\max$ in (8) from $c\in\{S+1,\ldots,S+U\}$ to $c\in\{1,\ldots,S+U\}$.

## 3 RELATED WORK

Despite attributed-based ZSL (Lampert et al., 2014; Changpinyo et al., 2016; Romeraparedes & Torr, 2015; Verma et al., 2018) have obtained exciting advances, the creation of attributes is usually realized by collecting lots of annotations from each of the seen and unseen images, which, however, is opposite to the motivation of less human annotation and the reality of few unseen samples. To remedy these drawbacks, text-based ZSL have been developed using easily accessible textural descriptions such as Wikipedia. Along this line, (Elhoseiny et al., 2013; 2017a; Ba et al., 2015; Qiao et al., 2016) learn explicit visual classifiers conditioned on the textual description with seen classes, further transferred to the unseen ones. Elhoseiny et al. (2017b) and Zhu et al. (2018) propose methods connecting text terms to the relevant visual parts extracted by a CNN-based detector without any part-text annotations. Though having achieved good performance on specific tasks, both of them heavily depend on visual part detector that has to be elaborately tuned for different classes manually. Meanwhile, all of them assume that each class is represented as a fixed point in semantic space, which does not adequately account for data variability (Akata et al., 2015), due to lacking proper probabilistic generative modeling for the data.

From the view of the model, the proposed VHEGAN is able to generate not only the visual images but also the documents with a shared probabilistic space, which can be also seen as a multimodal learning. In Srivastava & Salakhutdinov (2012; 2014), a deep Boltzmann machine is developed to model the joint generation of image feature and BOW tags. Moving beyond binary hidden units, Wang et al. (2018a) propose a multimodal Poisson gamma belief network, similar to the document decoder of VHEGAN. However, the Poisson likelihood with gamma link restricts its ability on describing low-level image features. In Reed et al. (2016) and Gomez et al. (2017), a convolutional GAN and a CNN model are used to focus on describing image from text and translating image into textual features in one direction, respectively. Both these two approaches are two-step models with the additional textual feature extraction realized by LSTM and "shallow" LDA, respectively, whereas the VHEGAN can realize joint optimization with hybrid SGMCMC/VHE/SGD algorithm. Besides, VHEGAN is able to realize multi-directed transformation, where ZSL can be seen as predicting the BOW textual features from the images.

In terms of conditional generative model, in order to use label to affect generative process (Kingma et al., 2014), or use observations of one domain to generate structured distributed output belonging to another (Sohn et al., 2015), conditional VAE, a conditional directed graphical model whose input observations modulate the prior on Gaussian latent variables to generate the outputs, is developed. VHE builds a variational posterior conditioned on images to approximate Gamma latent variables to generate the BOW vector of documents. To solve the problem of complete random generate process in GAN, conditional GANs (CGAN) (Mirza & Osindero, 2014; Reed et al., 2016) are also

Table 1: Accuracy (%) of zero-shot classification on CUB2011-hard, CUB2011-easy, and Flower Datasets. Note that some of them are attribute-based methods but applicable in our setting by replacing attribute vectors with text features (labeled by *), as discussed in Elhoseiny et al. (2017b). ZSLPP and GAZSL use a well-defined part detection features for images. All results on CUB2011 and the results of WAC on Flower come from Elhoseiny et al. (2017b); Qiao et al. (2016); Elhoseiny et al. (2017a), with all the others obtained by running the code provided by the original authors.

| Text-ZSL dataset | CUB2011-hard | CUB2011-easy | | Flower |
|---|---|---|---|---|
| Accuracy criterion | top-1 | top-1 | top-5 | top-1 |
| DNN-based (Ba et al., 2015) | – | 12.0 | 42.8 | – |
| WAC-Linear (Elhoseiny et al., 2013) | 5.0 | 27 | – | 5.9±1.48 |
| WAC-Kernel (Elhoseiny et al., 2017a) | 7.7 | 33.5 | – | 9.1±2.77 |
| ZSLNS (Qiao et al., 2016) | 7.3 | 29.1±0.28 | 61.8±0.22 | 8.7±2.46 |
| ESZSL* (Romeraparedes & Torr, 2015) | 7.4 | 28.5 | 59.9 | 8.6±2.53 |
| SynC* (Changpinyo et al., 2016) | 8.6 | 28.0 | 61.3 | 8.2±2.31 |
| SJE* (Akata et al., 2015) | – | 29.9 | – | – |
| ZSLPP (Elhoseiny et al., 2017b) | 9.7 | 37.2 | – | – |
| GAZSL (Zhu et al., 2018) | 10.3 | **43.7** | *65.24* | – |
| VHE-layer1 | 9.2±0.32 | 28.6±0.31 | 57.1±0.33 | 7.9±1.66 |
| VHE-layer2 | 12.4±0.26 | 32.5±0.28 | 61.2±0.24 | 8.5±1.60 |
| VHE-layer3 | 14.0±0.24 | 34.6±0.25 | 64.6±0.20 | 8.9±1.57 |
| VHEGAN-layer1 | 10.3±0.31 | 30.6±0.32 | 60.8±0.33 | 8.8±1.68 |
| VHEGAN-layer2 | 13.8±0.26 | 35.9±0.24 | 63.4±0.25 | 9.2±1.54 |
| VHEGAN-layer3 | **15.7±0.24** | 37.4±0.20 | **66.3±0.18** | **9.8±1.47** |

developed, where the conditions are labels, or attributes learned as point estimations separately with GAN, while VHEGAN learns a distributed condition jointly with a VAE model. Moreover, VHEGAN is similar with BiGAN (Donahue et al., 2017) and ALI (Dumoulin et al., 2017) since all of them learn a encoder that maps data into a latent space in GAN. Different from them that only use discriminator as the similarity measurement, VHEGAN fuse VHE for documents and GAN for images, making iteself realize not only an image-to-image "autoencoder" like BiGAN and ALI, but also an image-to-text and text-to-image "hetero-encoder".

## 4 EXPERIMENTS

The code for reproducible research will be made publicly available if the paper gets accepted. We evaluate the proposed models on two text-ZSL benchmark datasets: CUB2011 (Wah et al., 2011) and Oxford-flower (Nilsback & Zisserman, 2008). We consider VHE- and VHEGAN-layer1, VHE- and VHEGAN-layer2, and VHE- and VHEGAN-layer3, which use the Poisson GBNs with a single hidden layer of 256 hidden units, two hidden layers of size 256-128, and three hidden layers of size 256-128-64, respectively. CUB2011 consists of 11,788 images from 200 bird subspecies and Oxford-flower consists of 8189 images from 102 flower classes. The raw texts of all categories are provided in Elhoseiny et al. (2017a). After removing a standard list of stopwords, we use the 6000 most frequent words for CUB2011 and 3000 ones for Oxford-flower, respectively.

The 4096 dimensional image features, $f(x)$ in (3), from the fc1 layer of the pre-trained VGG16 network (Simonyan & Zisserman, 2015) are fed into the VHE image encoder, and the size of each generated RGB image is set as $64 \times 64$. We follow the same way in Elhoseiny et al. (2017a) to split the data. For Flower, five random splits were performed, in each of which $4/5$ of the classes are considered as "seen classes" for training and $1/5$ of the classes as "unseen classes" for testing. For CUB2011, there are two split settings: the hard one and the easy one. The hard one ensures that the bird subspecies belonging to the same super-category should belong to either the training split or test one without overlapping, referred to as CUB2011-hard. A recently used split setting (Qiao et al., 2016; Akata et al., 2015) is super-category split, where for each super-category, except for one subspecies that is left as unseen, all the other are used for training, referred to as CUB2011-easy.

### 4.1 ACCURACY FOR ZERO-SHOT LEARNING

We make comparison between a variety of methods that are suitable for text-ZSL, as summarized in Table 1, using Top-1 accuracy (some methods also provide Top-5 accuracy on CUB2011-easy) for

the unseen classes. Note except for the proposed VHEGAN that aims to find a shared semantically meaningful latent space between the image and text modalities, none of the other methods have generative models for both modalities, regardless of whether they learn a classifier or a distance metric in a latent space for ZSL. Table 1 shows that VHEGAN-layer3 clearly outperforms the state-of-the-art in terms of Top-1 accuracy on both the CUB2011-hard and Flower text-ZSL tasks, and is comparable to the second best on CUB2011-easy (it is also the best among all methods that have reported their Top-5 accuracies on CUB2011-easy). Note for CUB2011-easy, every unseen class has some corresponding seen classes under the same super-category, which makes the classification of surface or distance metric learned on the seen classes easier to generalize to the unseen ones. We also note that both GAZSL (Zhu et al., 2018) and ZSLPP (Elhoseiny et al., 2017b) rely on visual part detectection to extract image features, making their performance sensitive to the quality of the visual part detector that often has to be elaborately tuned for different classes and hence limiting their generalization ability, for example, the visual part detector for birds is not suitable for flowers.

For VHEs and VHEGANs with different network structures, Table 1 shows that given the same structure on the text decoder and image encoder, each VHEGAN consistently outperforms its VHE counterpart, suggesting the advantage of a joint generation of both the image and text modalities. It also shows that both VHEGAN and VHE have a clear improving trend as the Poisson GBN becomes deeper, suggesting the advantage of having a deep hierarchical representation for text generation.

## 4.2 ACCURACY FOR GENERALIZED ZERO-SHOT LEARNING

Under the GZSL setting, the testing set consists of examples from both the seen and unseen classes, with no prior information on the proportion of unseen examples. Following previous work on GZSL, we perform an 80/20 random split on the seen classes $p_{data}(\boldsymbol{x}, \boldsymbol{t})$ of CUB2011-easy to obtain $p_{data}(\boldsymbol{x}, \boldsymbol{t})^{train}$ for training, and $p_{data}(\boldsymbol{x}, \boldsymbol{t})^{test}$ with the unseen classes $p_{data}(\boldsymbol{x}, \boldsymbol{t})^{unseen}$ used to evaluate the GZSL performance with measures (Verma et al., 2018; Chao et al., 2016) denoted as

$Acc_s$: S → S+U: Average per-class classification accuracy on $p_{data}(\boldsymbol{x}, \boldsymbol{t})^{test}$;

$Acc_u$: U → S+U: Average per-class classification accuracy on $p_{data}(\boldsymbol{x}, \boldsymbol{t})^{unseen}$.

To mitigate the bias towards seen classes accuracy, we evaluate the harmonic mean of the above defined average top-1 accuracies as $H = (2 \times Acc_s \times Acc_u)/(Acc_s + Acc_u)$ (Verma et al., 2018).

Compiled in Table. 2 where the results are got from Verma et al. (2018) except ZSLPP, WAC, GAZSL and ours achieved by running provided code by 20 random splits on $p_{data}(\boldsymbol{x}, \boldsymbol{t})$ to achieve mean accuracy and error bar, it clearly demonstrates that our model can significantly mitigate the GZSL issue of the bias towards seen classes, which some compared approaches, such as SJE, ESZSL and SynC, tend to suffer from. Recaping the model architecture, SJE, ESZSL, SynC aim to learn a classifier based on class-specified textual description or attribute which are easier to bias to-

Table 2: Top-1 accuracy of GZSL on CUB2011-easy with different measures, where the abbr. of all methods are same with Table 1.

| Model | $Acc_u$ | $Acc_s$ | $H$ |
|---|---|---|---|
| WAC-Linear | 26.42±0.42 | 69.20±0.36 | 38.24±0.44 |
| WAC-kernel | 28.36±0.48 | 58.62±0.43 | 38.22±0.45 |
| SJE | 23.50 | 59.20 | 33.60 |
| ESZSL | 12.60 | 63.80 | 21.00 |
| SynC | 11.50 | 70.90 | 19.80 |
| ZSLPP | 23.60±0.35 | 70.12±0.24 | 35.32±0.40 |
| GAZSL | **33.61**±0.36 | 68.12±0.32 | **45.01**±0.33 |
| VHE-layer1 | 21.37±0.46 | 60.63±0.44 | 31.60±0.50 |
| VHE-layer2 | 26.52±0.33 | 65.48±0.28 | 37.75±0.34 |
| VHE-layer3 | 28.75±0.29 | 67.81±0.26 | 40.38±0.29 |
| VHEGAN-layer1 | 23.14±0.41 | 63.98±0.40 | 33.99±0.45 |
| VHEGAN-layer2 | 28.40±0.31 | 67.72±0.26 | 40.02±0.31 |
| VHEGAN-layer3 | 30.24±**0.28** | **70.31**±0.24 | 42.29±**0.27** |

wards to the seen class, while VHEGAN tries to find a latent space to represent the observations through generative models. Moreover, although ZSLPP and GAZSL use part detector learned with accurate part bonding-box, which is hard obtained in many ZSL cases, our best model VHEGAN-layer3 outperforms ZSLPP, and a little worse than GAZSL on $Acc_u$ and and $H$.

## 4.3 QUALITATIVE EVALUATION

In addition to quantitative evaluation on text-ZSL tasks, below we provide qualitative analysis to demonstrate the ability of VHEGAN in jointly modeling both the text and image modalities.

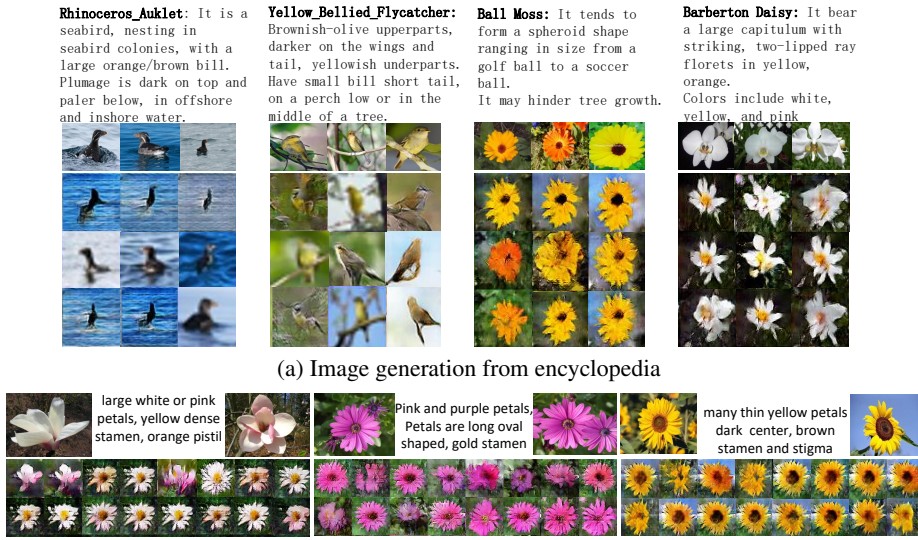

(a) Image generation from encyclopedia

(b) Image generation from textual caption

Figure 2: Class specific image generation based on the textual descriptions of unseen classes. (a) CUB2011-easy and Flower data, where the first row shows real samples, while the others generated images; (b) Flower data, where the top row shows the representative textual captions and two real images of the same class.

**Text to image generation:** Although VHEGAN does not have a text encoder to directly project a document to the shared latent space, given a document and a set of topics inferred during training, we use the upward-downward Gibbs sampler of Zhou et al. (2016) to draw $\{\boldsymbol{\theta}^{(l)}\}_{l=1,L}$ from its conditional posterior under the Poisson GBN, which are then fed into the GAN image generator to generate random images. Given example encyclopedia documents for the unseen classes, we follow this approach to generate random images.

We train VHEGAN on CUB2011-easy and then perform image generation given the textual discriptions of four different unseen classes. Comparing the generated images, as shown in the third to fifth rows of Fig. 2a, with their corresponding texts and example real images, as shown in the first and second rows of Fig. 2a, respectively, clearly suggest that the proposed VHEGAN successfully transfers the learned knowledge of the seen classes to the unseen ones, generating images semantically (visually) similar to their corresponding textual descriptions (real images).

We repeat the text-to-image generation experiment on Flower to generate random images given textual captions for three different unseen classes. As shown in Fig. 2b, VHEGAN successfully generates flower images that semantically match their corresponding textual descriptions and visually resemble their corresponding example real images.

**Image retrieval given a text query:** For image $\boldsymbol{x}_i$, via the text generator and image encoder , we generate a bag-of-words vector $\hat{\boldsymbol{t}}_i$ using Poisson mean $\boldsymbol{\Phi}^{(1)}\boldsymbol{\theta}^{(1)}$ as

$$\hat{\boldsymbol{t}}_i \,|\, \boldsymbol{\theta}_i \sim p(\boldsymbol{t}\,|\,\boldsymbol{\Phi}, \boldsymbol{\theta}_i), \ \boldsymbol{\theta}_i \,|\, \boldsymbol{x}_i \sim q_{\boldsymbol{\Omega}}(\boldsymbol{\theta}\,|\,\boldsymbol{\Phi}, \boldsymbol{x}_i). \tag{9}$$

Giving the textural description $\boldsymbol{t}$ of an unseen class as the text query, we retrieve the top five images ranked by the cosine distances between $\boldsymbol{t}$ and $\hat{\boldsymbol{t}}_i$. Shown in Fig. 3 are two example image retrieval results, which suggest that the retrieved images are semantically related to their text queries in colors, shapes, and locations. To be more clear, we use a colored dot on the top right corner of each image to denote its corresponding class, with the orange and red dots denoting the ground-truth classes associated with the left and right text queries, respectively. It is worth noting although some retrieved images among the top 5 are from different classes, they nicely match the corresponding text query, which explains why achieving high ZSL accuracy on Flower is not an easy task.

**Image to text:** The text-ZSL accuracies in Table 1 have already illustrated the effectiveness of VHEGAN in retrieving relevant textual descriptions given an image. With (9), we may generate a textual description given an image, as shown in Fig. 4, where the true and generated key words are displayed on the left and right of the input image, respectively. It is clear that VHEGAN successfully captures the flower colors, shapes, and locations to impute relevant key words given an input image.

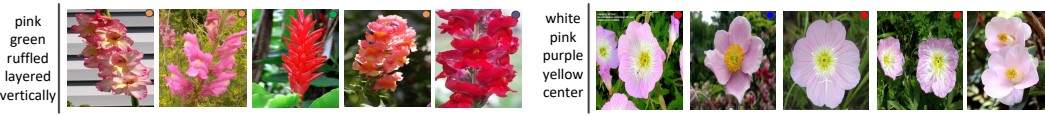

Figure 3: Top-5 retrieved images using the textual description of one class on Flower as the text query, where different colored dots at the top right corner denote different classes that the flowers belong to.

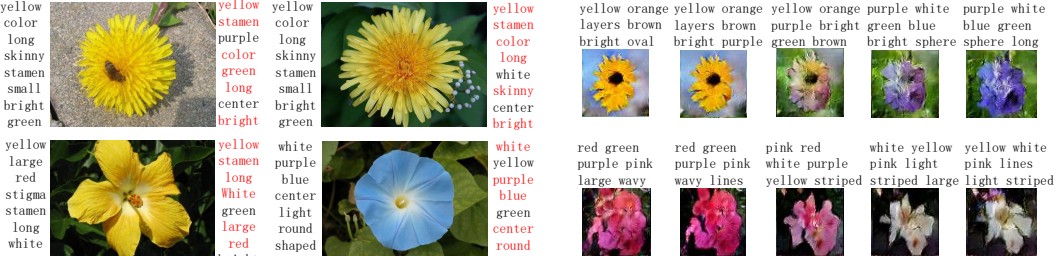

Figure 4: Image to textual tags on Flower with the red ones denoting the overlap with the ground-truth.

Figure 5: Documents interpolation to images on Flower

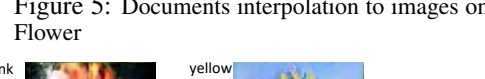
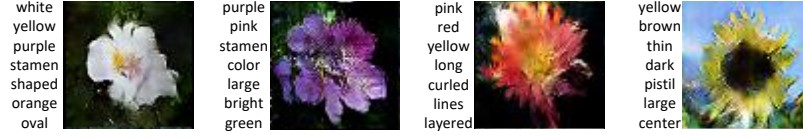

Figure 6: Image and document samples jointly generated from a three-layer VHEGAN.

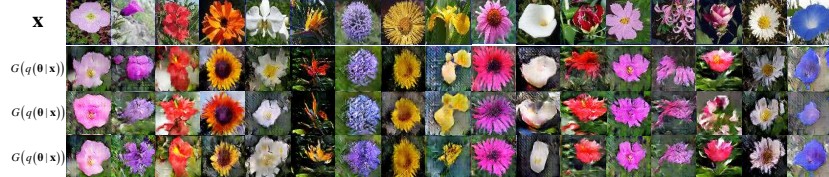

Figure 7: Image reconstruction with real data $x$ (the first row) and corresponding reconstructions $G\left(q_{\Omega}\left(\theta \mid \Phi, x\right)\right)$, where each rows represent different sampling from $q_{\Omega}\left(\theta\right)$.

**Latent space interpolation:** In order to understand the learned manifold in the latent space, given two texts $t_1$ and $t_2$, after drawing $\theta_1$ and $\theta_2$ from their own conditional posteriors using Gibbs sampling, we interpolate between $\theta_1$ and $\theta_2$ and use these interpolated latent variables to generate both the images via the GAN image generator and texts via the VHE text decoder. We show the results in Fig. 5, where the first and last columns show the true texts $t_1$ and $t_2$, and the images generated from $\theta_1$ and $\theta_2$, respectively, while the second to third columns show the generated texts and images from the interpolated $\theta$'s. The strong correspondences between the generated images and texts and smooth changes between adjacent columns suggest that the VHEGAN latent space is semantically meaningful for both the image and text modalities.

**Random text-image pairs generation:** Below we show how to generate data samples that contain both modalities. After training a 3-layer VHEGAN, following the data generation process of the VHE text decoder (*i.e.*, the Poisson GBN), given $\{\Phi^{(l)}\}_{l=1}^L$ and $r$, we first generate $\theta^{(L)} \sim \mathrm{Gam}\left(r, 1/s^{L+1}\right)$ and then downward propagate it through the Poisson GBN as in (1) to calculate the Poisson rates for all words using $\Phi^{(1)}\theta^{(1)}$. Given a random draw, the concatenation of $\{\theta^{(l)}\}_{l=1}^L$ is fed into the GAN image generator to generate a corresponding image. Shown in Fig. 6 are four random draws, for each of which we show its top seven words and its generated image, whose relationships are clearly interpretable, suggesting that VHEGAN is able to recode the key information of both modalities and the relationships between them.

**Image to image generation:**

We note for the VHEGAN, its VHE image encoder and GAN component together can also be viewed as an "autoencoding" GAN for images. Different from Dumoulin et al. (2017) and Donahue et al. (2017), the encoded latent space is affected not only by the GAN for the image modality, but also by the Poisson GBN for the text modailty. In Fig. 7, we present the real images $x$ and their VHEGAN

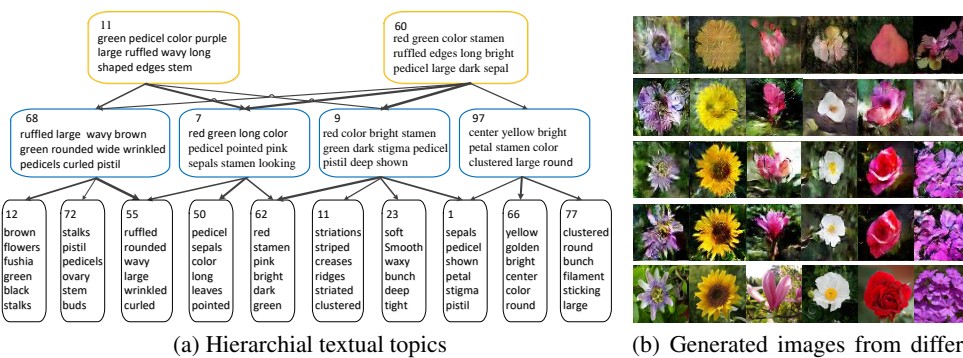

(a) Hierarchial textual topics

(b) Generated images from different layers

Figure 8: (a) is an example of learned hierarchical topics; (b) is the generated images, from the first to fourth rows, using the third, second, first, and the concatenation of all hidden layers fed into GAN, respectively. Each one in the bottom row is a real image belonging to the corresponding class of that column.

regenerations $G\left(q_{\boldsymbol{\Omega}}\left(\boldsymbol{\theta} \mid \boldsymbol{\Phi}, \boldsymbol{x}\right)\right)$, which suggest that the VHEGAN can use its GAN image generator to regenerate random images that more or less resemble the original real image fed into the VHE image encoder.

## 4.4 TOPIC HIERARCHY

The inferred topics at different layers and the inferred connection weights between the topics of adjacent layers are found to be highly interpretable. In particular, we can understand the meaning of each topic by projecting it back to the original data space via $\left[\prod_{t=1}^{l-1} \boldsymbol{\Phi}^{(t)}\right] \boldsymbol{\phi}_k^{(l)}$. We show in Fig. 8a a subnetwork, originating from units 11 and 60 of the top hidden layer, taken from the three-layer VHEGAN of size 256-128-64 inferred on Flower. The semantic meaning of each topic and the connection weights between the topics of adjacent layers are highly interpretable, where the topics describe very specific flower characteristics, such as colors, shapes, and parts, at the bottom layer, and become increasingly more general when moving upwards. In addition, after training the three-layer VHEGAN, given $\boldsymbol{\Phi}$, we modify it by feeding only one of the three hidden layers to the GAN image generator, and retrain the VHEGAN (only updating $G$, $D$ and $\boldsymbol{\Omega}$). For each column in Fig. 8b, conditioning on a textual description, we show the generated images, from the first to fourth rows, using the third, second, first, and the concatenation of all layers, respectively, as the latent space fed into the GAN image generator. An real example closest to the generated one (compared in the latent space) belonging to corresponding class of each column is also given at the bottom row. Examining Fig. 8b suggests that different hidden layers concentrate on somewhat different visual information and combining them lead to the best visual quality, a reason that by default all hidden layers are concatenated to fed into the GAN image generator.

## 5 CONCLUSION

To exploit and explore the relationships between the images and texts for text-based zero-shot learning (ZSL), we propose variational hetero-encoder (VHE) that encodes an image, via a Weibull distribution based inference network, to decode its textual description, via a deep probabilistic topic model. We introduce VHE randomized generative adversarial network (VHEGAN) to further strengthen the interaction between the image and text modalities in their shared latent space. With a stochastic-gradient MCMC algorithm to sample the global parameters of the VHE text decoder, a reparameterized Weibull distribution based variational encoder to approximate the posterior distribution of the local parameters of the VHE image encoder, and stochastic gradient descent (SGD) to estimate the parameters of the GAN image generator and discriminator, we develop a SG-MCMC/VHE/SGD hybrid inference algorithm to jointly train the components of VHEGAN. Both quantitative results on ZSL and generalized ZSL task and qualitative analysis suggest that VHEGAN is able to infer the parameters of probabilistic deep neural networks to extract and relate visual and linguistic concepts from the training data, and use the inferred networks to perform a wide variety of conditional or unconditional image/text generation/retrieval tasks.

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

---

**Algorithm 1** Jointly hybrid SGMCMC/VHE/SGD learning algorithm for VHEGAN

---

Set the number of class $C$ and the number of images belonging to a class $m$ in one mini-batch and the number of layer $L$.

Initialize encoder parameter $\boldsymbol{\Omega}$, topic parameters of DLDA $\{\boldsymbol{\Phi}^{(l)}\}_{1,L}$, generator $G$ and discriminator $D$.

**for** $iter = 1, 2, \cdots$ **do**

    Randomly select a mini-batch containing $C$ documents and its corresponding images $D = \{\boldsymbol{x}_i^c, \boldsymbol{t}_c\}_{i=1,c=1}^{m,C}$;

    Draw random noise $\left\{\varepsilon_i^{c,l}\right\}_{i=1,c=1,l=1}^{m,C,L}$ from uniform distribution;

    Calculate $\nabla_D L_I\left(D, G, \boldsymbol{\Omega}|\boldsymbol{x}\right)$;

    Calculate $\nabla_G L_I\left(D, G, \boldsymbol{\Omega}|\boldsymbol{x}\right)$;

    Calculate $\nabla_{\boldsymbol{\Omega}}\left[L_I\left(D, G, \boldsymbol{\Omega}|\boldsymbol{x}\right) - L_T(\boldsymbol{\Omega}, \{\boldsymbol{\Phi}^{(l)}\}_{l=1,L}|\boldsymbol{t})\right]$ by the aid of $\left\{\varepsilon_i^{c,l}\right\}_{i=1,c=1,l=1}^{m,C,L}$;

    Update $D$ as $D = D + \nabla_D L_I\left(D, G, \boldsymbol{\Omega}|\boldsymbol{x}\right)$;

    Update $G$ as $G = G - \nabla_G L_I\left(D, G, \boldsymbol{\Omega}|\boldsymbol{x}\right)$;

    Update $\boldsymbol{\Omega}$ as $\boldsymbol{\Omega} = \boldsymbol{\Omega} - \nabla_{\boldsymbol{\Omega}}\left[L_I\left(D, G, \boldsymbol{\Omega}|\boldsymbol{x}\right) - L_T(\boldsymbol{\Omega}, \{\boldsymbol{\Phi}^{(l)}\}_{l=1,L}|\boldsymbol{t})\right]$;

    Sample $\boldsymbol{\theta}_c^{\{l\}}$ from (3) via $\boldsymbol{\Omega}$ and $\{\boldsymbol{x}_i^c\}_{i=1,c=1}^{m,C}$ to update topics $\{\boldsymbol{\Phi}^{(l)}\}_{l=1}^{L}$ according to Cong et al. (2017);

**end for**

---

Liwei Wang, Alexander G Schwing, and Svetlana Lazebnik. Diverse and accurate image description using a variational auto-encoder with an additive gaussian encoding space. In *NIPS*, pp. 5756–5766, 2017.

Wenlin Wang, Yunchen Pu, Vinay Kumar Verma, Kai Fan, Yizhe Zhang, Changyou Chen, Piyush Rai, and Lawrence Carin. Zero-shot learning via class-conditioned deep generative models. In *AAAI*, 2018b.

Kelvin Xu, Jimmy Ba, Ryan Kiros, Kyunghyun Cho, Aaron C Courville, Ruslan Salakhudinov, Rich Zemel, and Yoshua Bengio. Show, attend and tell: Neural image caption generation with visual attention. In *ICML*, pp. 2048–2057, 2015.

Hao Zhang, Bo Chen, Dandan Guo, and Mingyuan Zhou. Whai: Weibull hybrid autoencoding inference for deep topic modeling. In *ICLR*, 2018.

Mingyuan Zhou and Lawrence Carin. Negative binomial process count and mixture modeling. *IEEE Trans. Pattern Anal. Mach. Intell.*, 37(2):307–320, 2015.

Mingyuan Zhou, Lauren Hannah, David Dunson, and Lawrence Carin. Beta-negative binomial process and Poisson factor analysis. In *AISTATS*, pp. 1462–1471, 2012.

Mingyuan Zhou, Yulai Cong, and Bo Chen. The Poisson Gamma belief network. In *NIPS*, pp. 3043–3051, 2015.

Mingyuan Zhou, Yulai Cong, and Bo Chen. Augmentable gamma belief networks. *J. Mach. Learn. Res.*, 17(163):1–44, 2016.

Yizhe Zhu, Mohamed Elhoseiny, Bingchen Liu, Xi Peng, and Ahmed M Elgammal. A generative adversarial approach for zero-shot learning from noisy texts. In *CVPR*, 2018.

## A    APPENDIX: HYBRID SGMCMC/VHE/SGD ALGORITHM FOR VHEGAN

We give a detailed hybrid SGMCMC/SGD algorithm for VHEGAN in this section, to realize learning end-to-end.

# B APPENDIX: MODEL ARCHITECTURE

In Fig. 1, we give a simple model structure of VHEGAN. In this appendix section, we give more detailed model architecture for all of three parts (image encoder named WUDVE, image decoder named Poission GBN, and image decoder GAN) of VHEGAN. The input of the VHEGAN is abbreviated as $\{x, t\}$ pair, which represents images and texts, respectively.

## B.1 ARCHITECTURE OF POISSION GBN (IMAGE DECODER)

For Poission GBN, only the number of topics in different layers should be set. In our experiment's setting, we set the number of topics from the first to the third layer to 256, 128 and 64, respectively.

## B.2 ARCHITECTURE OF WUDVE (IMAGE ENCODER)

Image encoder $x \to h^{(l)}, l = 1, 2, 3$

---

$x \to h^{(1)}$

$1\times$ { fully-connected layer with 256 units and Leaky-RELU activation.}

$h^{(1)} \to h^{(2)}$

$1\times$ { fully-connected layer with 128 units and Leaky-RELU activation.}

$h^{(2)} \to h^{(3)}$

$1\times$ { fully-connected layer with 64 units and Leaky-RELU activation.}

---

Weibull variational posterior $q(\boldsymbol{\theta}^{(l)}|h^{(l)})$ as shown in Fig. :

$h^{(l)} \to k_{\boldsymbol{\theta}^{(l)}}, \boldsymbol{\lambda}_{\boldsymbol{\theta}^{(l)}}; l = 1, 2, 3$

---

$h^{(1)} \to k_{\boldsymbol{\theta}^{(1)}}, \boldsymbol{\lambda}_{\boldsymbol{\theta}^{(1)}}$

$1\times$ { fully-connected layer with 256 units to $k_{\boldsymbol{\theta}^{(1)}}$ and $\boldsymbol{\lambda}_{\boldsymbol{\theta}^{(1)}}$.}

$h^{(2)} \to k_{\boldsymbol{\theta}^{(2)}}, \boldsymbol{\lambda}_{\boldsymbol{\theta}^{(2)}}$

$1\times$ { fully-connected layer with 128 units to $k_{\boldsymbol{\theta}^{(2)}}$ and $\boldsymbol{\lambda}_{\boldsymbol{\theta}^{(2)}}$.}

$h^{(3)} \to k_{\boldsymbol{\theta}^{(3)}}, \boldsymbol{\lambda}_{\boldsymbol{\theta}^{(3)}}$

$1\times$ { fully-connected layer with 64 units to $k_{\boldsymbol{\theta}^{(3)}}$ and $\boldsymbol{\lambda}_{\boldsymbol{\theta}^{(3)}}$.}

## B.3 ARCHITECTURE OF GAN

Image Generation Network shown in Fig. 10 : $\boldsymbol{\theta}^{(l)} \to \hat{x}; l = 1, 2, 3$

---

Concatenate $\boldsymbol{\theta}^{(l)}$ on feature's dimension, $l = 1, 2, 3$, to get $\boldsymbol{\theta}$

$1\times$ { fully-connected layer with $4 \times 4 \times 64 \times 4$ units, RELU activation and batch normalization.}

$1\times$ { deconv2d 256 feature maps with $5 \times 5$ kernels, stride 2, RELU activation.}

$1\times$ { deconv2d 128 feature maps with $5 \times 5$ kernels, stride 2, RELU activation.}

$1\times$ { deconv2d 64 feature maps with $5 \times 5$ kernels, stride 2, RELU activation.}

$1\times$ { deconv2d 256 feature maps with $5 \times 5$ kernels, stride 2.}

---

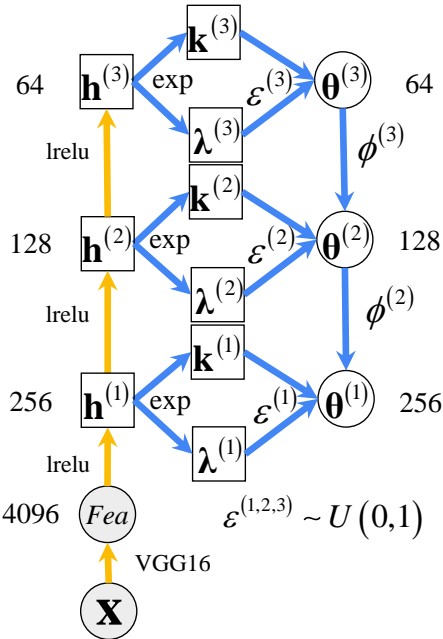

Figure 9: The architecture of WUDVE in VHE and VHEGAN.

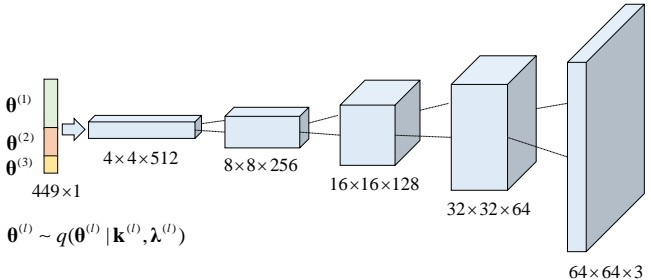

Figure 10: Generative model in GAN of VHEGAN

Discriminator Network shown in Fig. 11: $\hat{x}, x \rightarrow \{0/1\}$

---

Taking $x$ for example, $\hat{x}$ is the same.

$1\times$ { conv2d 64 feature maps with $5 \times 5$ kernels, stride 2.}

$1\times$ { conv2d 128 feature maps with $5 \times 5$ kernels, stride 2, Leaky-RELU activation.}

$1\times$ { conv2d 256 feature maps with $5 \times 5$ kernels, stride 2, Leaky-RELU activation.}

$1\times$ { conv2d 512 feature maps with $5 \times 5$ kernels, stride 2, Leaky-RELU activation.}

$1\times$ { fully-connected layer with 1 units.}

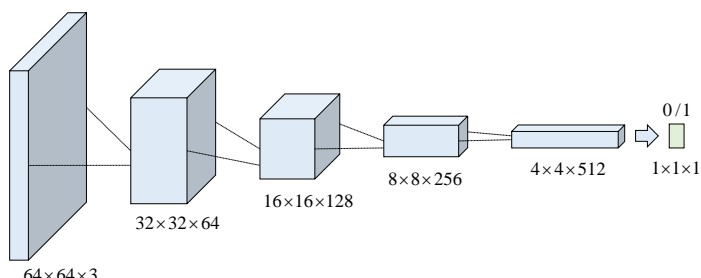

Figure 11: Discriminant model in GAN of VHEGAN

