# OpenReview forum: "VHEGAN: Variational Hetero-Encoder Randomized GAN for Zero-Shot Learning"
_ICLR.cc/2019/Conference_

### Official Review · AnonReviewer2 · 2018-11-01
**Interesting paper, borderline results.**

**Rating:** 5
**Confidence:** 5

**Review:**

Paper Summary: This paper studies the zero-shot learning problem with deep generative models. More specifically, it proposed a hybrid framework that combines VAEs (more precisely, the variational hetero-encoder or VHE) and GANs all together. The entire model is composed of an image encoder (Weibull upward-downward variational encoder), a text decoder (Poisson Gamma belief network), and an image generator (generative adversarial network). Once learned, the generative models can be directly used for zero-shot classification and various image generation applications. In the experiments, two benchmark datasets CUB and Oxford-Flowers are used.

==
Novelty/Significance:
Zero-shot learning is a challenging task and he main motivation of the paper (using generative model) is interesting. The text representation in the paper is simply bag-of-words which limits the application to some extent. In a broader context, image captioning using generative model seems quite relevant.

Diverse and Accurate Image Description Using a Variational Auto-Encoder with an Additive Gaussian Encoding Space, Wang et al. In NIPS 2017.

==
Quality:
Overall, reviewer feels this is a very interesting work. However, the results from the paper is quite mixed. It is not yet convincing whether the proposed approach is the state-of-the-art in zero-shot learning or text-to-image generation.

First, this paper demonstrates the power of generative models in text-to-image generation and other applications. However, reviewer feels that the zero-shot classification result is weak. In Table 1 and Table 2, it seems that GAZSL (Zhu et al. 2018) outperforms the proposed approach.

Q1: In Table 2, is it possible to report the top-5 accuracy on CUB-easy and top-1 accuracy on Oxford-Flower dataset? Otherwise, it is not very convincing that proposed approach is better than the state-of-the-art approach GAZSL.

Second, the text-to-image generation results look reasonably good. But the resolution and quality of generated images are far from state-of-the-art. One suggestion is to train the VHE model with an improved image generator.

StackGAN: Text to Photo-realistic Image Synthesis with Stacked Generative Adversarial Networks, Zhang et al. In CVPR 2017.

AttnGAN: Fine-Grained Text to Image Generation with Attentional Generative Adversarial Networks, Xu et al. In CVPR 2018.

Also, reviewer would expect to see an improved image generator can lead to a better ZSL performance.

Typo: In the title: Zero-Short → Zero-Shot.

---

> ### Author Response · Authors · 2018-11-25
> **Response to Reviewer 2**
>
> We thank Reviewer 2 for his/her comments and suggestions.
>
> Q1: The text representation in the paper is simply bag-of-words which limits the application to some extent. In a broader context, image captioning using generative model seems quite relevant.
>
> A1: Image captioning often aims at generating a sequential text describing a certain image, which, however, might not be suitable for the ZSL classification task. First, a text representation is used to define a class of images (not just a single image). In such circumstance, compared with the sequential information, the key words are more general and effective to define a certain image class. Besides, the text representation for a class of images varies from a few words (e.g., a sentence or several tags) to hundreds of sentences (e.g., in encyclopedia), whose key words can be captured more robustly by PGBN than sequential models. The experimental results also illustrate that VHEGAN is able to capture semantically important words from both a long document and a short sentence, as illustrated in Figs. 2-6 in the revised manuscript.
>
> Q2: In Table 1, is it possible to report the top-5 accuracy on CUB-easy and top-1 accuracy on Oxford-Flower dataset? Otherwise, it is not very convincing that proposed approach is better than the state-of-the-art approach GAZSL. Reviewer feels that the zero-shot classification result is weak. In Table 1 and Table 2, it seems that GAZSL (Zhu et al. 2018) outperforms the proposed approach.
>
> A2: We only report top-1 accuracy on CUB-easy for GAZSL in Table 1 since the authors in Zhu et al. (2018) only list top-1 accuracy. With the code provided for Zhu et. (2018), we run it by ourselves and achieve 65.24% accuracy, a little lower than VHEGAN-layer3, as shown in the revised manuscript.
>
> From Table 1, we can see that GAZSL performs better only on CUB-easy and worse on both CUB-hard and Flower than VHEGAN does. Besides, as discussed in the manuscript, CUB-hard and Flower are more challenging datasets, which illustrates the generalization of VHEGAN. What’s more, lower error bars are achieved by our proposed models, demonstrating the robustness and effectiveness of the proposed posterior representation at multiple stochastic layers.
>
> In addition, although GAZSL exhibits higher accuracy on some metrics, it relies on an extra visual part detection that needs additional resources and elaborate tuning for different classes. As a result, as the original GAZSL only designs part detector for Birds, it is not suitable to perform ZSL on Flower. This problem may limit GAZSL in practical applications which contain many different types of objects.
>
> Q3: the text-to-image generation results look reasonably good. But the resolution and quality of generated images are far from state-of-the-art. One suggestion is to train the VHE model with an improved image generator.
>
> A3: Thank you for your suggestion. The resolution and quality of the generated images are affected by the type of GAN used in VHEGAN. Compared with generating high-quality images, for the ZSL task the paper is focused on, extracting image features to help the PGBN to learn a better latent space for the classification task is more important. Thus, we propose a framework to combine VAE-like model with GAN through a shared latent space, in which many types of GAN can be selected. According to the experiments, using DCGAN as image generator can realize a satisfactory classification results, which validates the effectiveness of our combination of PGBN and GAN for ZSL task. We leave the use of more sophisticated GAN in VHEGAN for future research.
>
> Q4: One suggestion is to train the VHE model with an improved image generator.
> StackGAN: Text to Photo-realistic Image Synthesis with Stacked Generative Adversarial Networks, Zhang et al. In CVPR 2017.
> AttnGAN: Fine-Grained Text to Image Generation with Attentional Generative Adversarial Networks, Xu et al. In CVPR 2018.
> Also, reviewer would expect to see an improved image generator can lead to a better ZSL performance.
>
> A4: Thank you for suggesting these two relevant papers. Due to the time constraint, we were not able to update our experimental results by replacing DCGAN with more sophisticated GANs. If the paper gets accepted, we will cite these two papers and try to use StackGAN and AttnGAN to replace DCGAN and see whether we can get improved results.

---

### Official Review · AnonReviewer3 · 2018-11-02
**[Review] VHEGAN: Variational Hetero-Encoder Randomized GAN for Zero-Short Learning**

**Rating:** 5
**Confidence:** 5

**Review:**

[Paper Summary]
This work suggests a new model incorporating deep topic model (text decoder),  VHE (image encoder), and GAN. The topic model and the VHE shares the topic parameters, and the GAN generate an image regarding the topic. Then, for ZSL, the image is encoded to corresponding topic parameters, and the parameter can tell which text description (unseen) is matched with the highest probability. GAN model is used to generate an image given the topic distribution. During the training of the GAN, the VHE and topic model is jointly trained and can enhance the ZSL performance marginally.

[pros]
- This work successfully incorporated the topic model and image encoding/decoding. All the individual parts are already given, but I think incorporating them in terms of a unified probabilistic model is also meaningful for this field.
- This work shows superior performance on the image to text ZSL problem.
- This work mapped the text to image, image to text mapping in a generative manner.

[cons]
- The problem is only valid when the unseen class distribution is very similar to the given classes. For example, the text description of unseen classes should be well represented to the topics from seen classes.
- It is doubtful that this corresponds to the term zero-shot learning; dealing with the case that the unseen class and the seen class are notably different from each other.
- Similarly, GAN learns images from the seen classes, and by nature, GAN would not generate the proper images of the unseen class if the image distribution of the unseen class is different to the already seen class. In the paper, the classes are very similar to each other (birds, flowers) and that would be the reason GAN worked in this model.
- (minor) The likelihood of the text (image) given topic should be provided and compared to the existing models.


[Summary]
- The reviewer is personally interested in the proposal of the work, but concern that ZSL is difficult to be the main target of the paper because the model can only deal with the classes with (very) similar semantics, and this is the main reason for the rating. The testing with more diverse class should be given, or solid explanation of the mentioned problem would be required.

---

> ### Author Response · Authors · 2018-11-25
> **Response to Reviewer 3**
>
> Q1: It is doubtful that this corresponds to the term zero-shot learning; dealing with the case that the unseen class and the seen class are notably different from each other
>
> A1: Most of existing ZSL algorithms try to learn a mapping from images to texts or attributes on seen classes, and fix the mapping for the unseen class to obtain the textual descriptions, which implies the assumptions that the mapping extracts the similar features in both seen and unseen images, and that the mapping approximates the similar features in both the seen and unseen textual descriptions. In other words, they are dealing ZSL as the case that the seen and unseen classes are related in some feature space to some extent, though they have not specified the meaning of the shared features explicitly. In our work, we give the ZSL task an interpretable shared latent space, connecting the relationships between the images structures and the key words, whose effectiveness is validated on benchmark datasets following the widely used experimental protocols for text-based ZSL. Just as you mentioned, the less the relationship between the seen and unseen classes, the more challenging the task is. Compared with CUB2011-easy dataset, the Cub2011-hard and Flower datasets are more challenging, where the proposed VHEGAN remains the best.
>
> Q2: The problem is only valid when the unseen class distribution is very similar to the given classes. For example, the text description of unseen classes should be well represented to the topics from seen classes.
>
> A2: Although the seen and unseen classes are related in some feature space, which is hard to be defined, as we mentioned in A1, the entire class distributions are not ‘very’ similar, since they are different classes after all. Certainly, as discussed before, this type of similar image-text relationships is a basic assumption in ZSL and we follow the widely used experimental protocols. What’s more, besides representing every specific text description of each class, the more important effect of explicitly learned topics is to well express the shared information between the seen and unseen classes, making the knowledge efficiently transfer from the seen to unseen classes.
>
> Q3: Similarly, GAN learns images from the seen classes, and by nature, GAN would not generate the proper images of the unseen class if the image distribution of the unseen class is different to the already seen class. In the paper, the classes are very similar to each other (birds, flowers) and that would be the reason GAN worked in this model.
>
> A3: According to the available ZSL protocols, the seen and unseen classes are related more or less. Besides, compared with the original GAN with an uninformative noise as the source distribution, in our model, the source distribution is related to the image distribution in semantic, which makes it possible to generate different images distributions from that of the seen classes.

---

### Official Review · AnonReviewer1 · 2018-11-06
**Extensive experiments but limited novelty**

**Rating:** 5
**Confidence:** 4

**Review:**

This paper developed a generative model to perform simultaneous embedding/generation of images/texts, with application to zero-shot learning. The experiments are extensive.

The novelty of this work is lacking.
The proposed method consists of a bag of existing models proposed by previous works.
But why using a certain model is not justified or explanined.
For example, for image generation, why use GAN instead of VAE.
For text encoding, why use PGBN, instead of recurrent VAE.

The method seems deteched from the problem of ZSL.
Throughout the paper, the authors mostly talk about how to perform joint embedding of texts and images. They give ZSL a touch, but as a side thing.
I would suggest the authors to position this work as a text/iamge embedding/generation work. Then use ZSL as an application.

The writing needs to be significantly improved. In the first paragraph describing the problem of ZSL, the authors end up with talking about the evaluation metric of ZSL.

---

> ### Author Response · Authors · 2018-11-25
> **Response to Reviewer 1**
>
> We thank Reviewer 1 for his/her comments and suggestions.
>
> For the text-based ZSL task, the key to success is to explore the relationship between all the images in a class (not a single image) and its class-specific textual description. For this purpose, a Variational Hetero-Encoder is proposed for this task, exhibiting good performance with a shared latent space extracted from these two modes. Though PGBN and GAN have been used to represent text and image, respectively, they have not been successfully combined to find the relationship between these two modes and the ZSL task. Compared with PGBN, an encoder network is used perform end-to-end inference, serving as the source distribution for GAN, which further reinforce the effect of text on image generation.
>
> Using PGBN instead of recurrent VAE and using GAN instead of VAE are due to the following reasons:
>
> First, it has been observed that the fully-connected VAE is not expressive enough for 64*64 RGB images and the deconvolutional VAE is hard to converge, which can be solved by GAN.
>
> Second, our task focuses on ZSL classification. Although a sequential description could be excellent at defining a specific individual image, the key words are more effective to define a class of images (not a specific image). In addition, recurrent VAE often fails for long sequential texts, e.g., the description of an image class in encyclopedia often consists of thousands of words. Therefore, we extract bag-of-words features and use PGBN for textual generation, which is more suitable for the ZSL task. Indeed, we have clearly identified semantic connections between the images and some key words in the class-specific textual descriptions, which have been effectively captured by our model, as illustrated in Figs. 2-6 in the revised manuscript. Following your suggestion, we have added more discussions and illustrations (highlighted with italic text style).
>
> Thank you for your suggestion that we can position this work as a text/image embedding/generation work, and then use ZSL as one of the applications. One reason we are focusing on ZSL in this paper is because ZSL is an application that we can provide rigorous quantitative comparison with previous work on this task. We are extending the proposed work to more applications and will report our findings in our future work.
>
> As for the writing, in particular the first paragraph, we will make careful changes if the paper gets accepted.

---

### Author Response · Authors · 2018-10-03
**Typo correction**

Our submission has a typo that will be corrected: "zero-short learning" shall be changed to "zero-shot learning"

---

### Author Response · Authors · 2018-11-27
**Clarifications of our main contributions**

Dear Reviewers,

Thank you for your constructive feedback. We have added more discussions in our revision to explain why we choose certain components to construct the proposed VHEGAN.

As also noted in your reviews, VHEGAN is not limited to the ZSL application. We choose to focus on ZSL mainly because 1) it is the original motivation for us to develop VHEGAN, 2) it allows us to provide not only interpretable visualization on the inferred latent space, but also objective quantitative comparison to verify its effectiveness in a concrete setting. Having excellent performance in the challenging text based ZSL tasks, we believe VHEGAN can also be generated to many other different machine learning tasks, and we are working on these extensions.

While there are several concerns/suggestions on the choices of the modeling components of VHEGAN, we'd like to emphasize that the VHEGAN framework and the idea of randomizing the noise of the GAN generator with another deep generative model are our key contributions.

The VHEGAN framework is very flexible. As also noted in your reviews, one may easily substitute its current VHE decoder (Poisson GBN), VHE encoder (Weibull upward-downward variational autoencoder), DCGAN discriminator, and DCGAN generator with different modules, either for improved ZSL or image/text generation performance, or for other applications different from the ZSL task focused in this paper. We are incorporating your suggestions to run more experiments (e.g., replacing DCGAN with StachGAN/AttGAN), and will report updated experimental results in our next revision if the paper gets accepted.

---

### Meta-Review · Area_Chair1 · 2018-12-11

**Confidence:** 4
**Recommendation:** Reject

**Metareview:**

The paper received borderline ratings due to concerns regarding novelty and experimental results/settings (e.g. zero shot learning). On my side, I believe that the proposed method would need more evaluations on other benchmarks (e.g., SUN, AWA1 and AWA2) for both ZSL and GZSL settings to make the results more convincing. Overall, none of the reviewers championed this paper and I would recommend weak rejection.